# ACE2-Independent Alternative Receptors for SARS-CoV-2

**DOI:** 10.3390/v14112535

**Published:** 2022-11-16

**Authors:** Suhyeon Lim, Monica Zhang, Theresa L. Chang

**Affiliations:** 1Public Health Research Institute, New Jersey Medical School, Rutgers, The State University of New Jersey, Newark, NJ 07103, USA; 2Department of Microbiology, Biochemistry and Molecular Genetics, New Jersey Medical School, Rutgers, The State University of New Jersey, Newark, NJ 07103, USA

**Keywords:** SARS-CoV-2, alternative receptors

## Abstract

Severe acute respiratory syndrome-related coronavirus (SARS-CoV-2), the causative agent of coronavirus disease 2019 (COVID-19), is highly contagious and remains a major public health challenge despite the availability of effective vaccines. SARS-CoV-2 enters cells through the binding of its spike receptor-binding domain (RBD) to the human angiotensin-converting enzyme 2 (ACE2) receptor in concert with accessory receptors/molecules that facilitate viral attachment, internalization, and fusion. Although ACE2 plays a critical role in SARS-CoV-2 replication, its expression profiles are not completely associated with infection patterns, immune responses, and clinical manifestations. Additionally, SARS-CoV-2 infects cells that lack ACE2, and the infection is resistant to monoclonal antibodies against spike RBD in vitro, indicating that some human cells possess ACE2-independent alternative receptors, which can mediate SARS-CoV-2 entry. Here, we discuss these alternative receptors and their interactions with SARS-CoV-2 components for ACE2-independent viral entry. These receptors include CD147, AXL, CD209L/L-SIGN/CLEC4M, CD209/DC-SIGN/CLEC4L, CLEC4G/LSECtin, ASGR1/CLEC4H1, LDLRAD3, TMEM30A, and KREMEN1. Most of these receptors are known to be involved in the entry of other viruses and to modulate cellular functions and immune responses. The SARS-CoV-2 omicron variant exhibits altered cell tropism and an associated change in the cell entry pathway, indicating that emerging variants may use alternative receptors to escape the immune pressure against ACE2-dependent viral entry provided by vaccination against RBD. Understanding the role of ACE2-independent alternative receptors in SARS-CoV-2 viral entry and pathogenesis may provide avenues for the prevention of infection by SARS-CoV-2 variants and for the treatment of COVID-19.

## 1. Introduction

The coronavirus disease 2019 (COVID-19), caused by severe acute respiratory syndrome-related coronavirus (SARS-CoV-2), has remained a major challenge for public health since the first case was reported in December 2019. Although SARS-CoV-2 vaccines are widely available, fully vaccinated people are susceptible to variants [1,2,3,4]. The main SARS-CoV-2 entry is mediated by the binding of the SARS-CoV-2 spike protein (RBD) to the human angiotensin-converting enzyme 2 (ACE2) receptor on the cell surface. The molecular mechanism of ACE2-mediated viral entry has been summarized in an excellent review [5]. This entry process involves viral attachment, the proteolytic cleavage of SARS-CoV-2 spike proteins into S1 and S2 fragments, endocytosis, and membrane fusion [5]. The S1 protein includes the N-terminal (NTD) domain and RBD, whereas the S2 protein promotes membrane fusion. The RBD is the primary target for anti-spike neutralization antibodies in response to infection and vaccination and has been used for therapeutics [5,6]. In addition to ACE2 receptors, some human cells possess ACE2-dependent accessory receptors to promote SARS-CoV-2 entry. These accessory receptors/molecules include furin [7,8], transmembrane serine protease 2 (TMPRSS2) and TMPRSS4 [9,10], trypsin [11], cathepsins [12,13], neuropilin-1 [14,15], sialic acid-containing glycolipids [16], vimentin [17,18], heparan sulfate [19], and phosphatidylserine receptor [20], all of which can promote SARS-CoV-2 viral entry in an ACE2-dependent manner. Furthermore, the IgG receptors FcgRIIA and FcgRIIIA contribute to the antibody-dependent enhancement of SARS-CoV-2 infection of cells in the presence of ACE2 [21]. Bioinformatics analysis suggests that molecules such as glucose-regulated protein 78 (GRP78) [22,23] or angiotensin II receptor type 2 (AGTR2) [24] can interact with SARS-CoV-2 spike proteins and may promote viral entry [25]. While ACE2-expressing cells support robust SARS-CoV-2 viral replication [26], ACE2 expression profiles are not completely associated with clinical manifestation or immune responses [26,27]. In the lung, for example, ACE2 expression is low abundance and limited to type II alveolar cells (AT2) and ciliated cells [28], yet lung pathology is not limited to these cells. Importantly, SARS-CoV-2 infects organs or cells that do not express ACE2, indicating the involvement of alternative receptors for SARS-CoV-2 [23,25,29,30,31,32]. The ACE2-independent SARS-CoV-2 entry can be resistant to antibodies targeting the spike RBD [30]. The fact that the highly transmissible Omicron BA.1 variant has evolved to be less dependent on TMPRSS2, resulting in the use of different entry pathways, a cell tropism shift, and altered pathogenesis [33,34,35,36] suggests that alternative receptors may also contribute to viral evolution and immune escape. Thus, a better understanding of the role of alternative receptors for SARS-CoV-2 is critical for developing anti-viral therapeutics and strategies to dampen virus-mediated immune activation and disease outcomes. In this review, we summarize current knowledge on ACE2-independent alternative receptors for SARS-CoV-2 (Table 1). Several alternative receptors are also involved in the entry of other viruses (Table 2), which may have implications for pathogenesis in patients with co-infection. 

## 2. CD147

CD147, known as *basigin* or extracellular matrix metalloproteinase inducer (EMMPRIN), acts as an alternative receptor for SARS-CoV-2 entry into cells with low or undetectable ACE2 expression [30,39]. CD147 is expressed in epithelial, neuronal, and myeloid cells, as well as lymphocytes, and is distributed among various tissues, including the brain, gastrointestinal (GI) tract, and reproductive tissues [45,46]. CD147 is elevated in cancer tissues and is involved in modulating the tumor microenvironment and cancer progression [47,48,49,50,51,52,53]. Its level is higher in obese diabetic adults [54], which may contribute to a higher risk for severe COVID-19.

**Table 2 viruses-14-02535-t002:** Involvement of ACE2-indepenent alternative receptors in viruses other than SARS-CoV-2.

Receptor	Viruses	Reference
CD147	HIV, HBV, HCV, measles virus, CMV, KSHV, and SARS-CoV	[55,56,57,58,59,60,61,62,63,64]
AXL	Dengue, zika, Ebola, Lassa, Marburg virus, Hantaan virus, and Andes virus	[65,66,67,68,69,70,71]
CD209L and CD209	Sindbis virus, Ebola virus, Japanese encephalitis virus, HIV, HCV, influenza A virus, SARS-CoV	[71,72,73,74,75,76,77,78,79]
CLEC4G/LSECtin	Ebloa, Lassa, Marburg virus, filovirus Japanese encephalitis virus, SARS-CoV	[71,74,80,81,82,83]
KREMEN1	Coxsackievirus A10 and other human type A enteroviruses	[84,85,86]
ASGR1/CLEC4H1	Hepatitis E virus	[87]
LDLRAD3	Venezuelan equine encephalitis virus	[88]
TMEM30A/CD50A	Lujo virus	[89]

CD147 is a functional receptor for various pathogens, including measles, human immunodeficiency virus (HIV), hepatitis B virus (HBV), hepatitis C virus (HCV), SARS-CoV, Kaposi’s sarcoma-associated herpesvirus (KSHV), and *Plasmodium falciparum* [55,56,57,58,59,60,61,62,90] and enhances HIV infectivity in a viral cyclophilin A-dependent manner [63]. The data regarding the role of CD147 as a receptor for SARS-CoV-2 entry has been inconsistent [30,37,39], possibly due to the choice of cell lines with different ACE2 abundance, the design and preparation of spike proteins, and the assay systems. Using loss- or gain-of-function assays, Wang et al. showed that CD147 interacts with spike RBD proteins and mediates SARS-CoV-2 viral entry into cells with or without ACE2 [39]. Mepolizumab, a humanized CD147-neutralizing mAb, inhibits SARS-CoV-2 replication in vitro [39]. SARS-CoV-2 infection and pathology are found in human CD147 knock-in mice (C57BL/6J or NOD *scid* IL2Rgamma^null^) [39,91]. In contrast, Shilts et al. did not observe the binding of full-length spike protein to CD147 [37], and knocking down CD147 in Calu-3 cells, which express high levels of ACE2, did not impact the susceptibility of cells to SARS-CoV-2 infection [37]. Similarly, Ragotte et al. did not detect the binding of CD147 (expressed in bacteria) to either spike full-length or RBD proteins, and polyclonal anti-CD147 abs had no effect on SARS-CoV-2 infection of Vero cells, which also express highly abundant ACE2 [38]. We have previously shown that mAb targeting the spike RBD domain inhibited SARS-CoV-2 entry in HeLa-ACE2 cells but did not block the CD147-mediated viral entry into lung epithelial A459 cells, which express low levels of ACE2 mRNA, indicating that the RBD is not involved in CD147-mediated viral entry [30]. Anti-CD147 mAb and CD147 knockdown suppressed viral infection, indicating a role of CD147 in viral entry into A459 cells. Taken together, it appears that CD147 acts as an alternative entry receptor in cells with no or low abundance of ACE2, but its receptor function is not apparent in cells with high levels of ACE2 (Figure 1). The data are inconsistent regarding the role of spike RBD domain for CD147-mediated viral entry, and the molecular mechanism by which CD147-mediates SARS-CoV-2 entry remains to be defined. CD147 is involved in macropinocytosis [92], which is an actin-mediated, clathrin-independent endocytic process important for viral entry [93]. CD147 has been shown to promote the entry of pentamer-expressing human cytomegalovirus (CMV) into epithelial and endothelial cells through macropinocytosis [64]. Thus, the determination of the role of CD147 in SARS-CoV-2 viral entry via macropinocytosis in cells with a low abundance of ACE2 may provide insight into SARS-CoV-2 pathogenesis.

## 3. AXL

AXL (from the Greek *anexelekto* or uncontrolled) is a tyrosine-protein kinase receptor, initially named UFO for its unidentified function as a protein with oncogenic potential [94,95]. AXL, a member of the TAM receptors (with Tyro3 and Mer) and its ligand GAS6 modulate innate immune responses and play a critical role in cancer progression and resistance to targeted therapies [95,96,97]. AXL is expressed in CD34+ progenitors, marrow stromal cells, peripheral monocytes, and bronchial cells but not in lymphocytes or granulocytes [98,99]. Phorbol ester and IFNα induce AXL expression in K562 chronic myeloid leukemia cells and monocytes, respectively [99]. AXL is broadly expressed in various tissues, including the respiratory system, GI tract, reproductive tissues, and muscles, and at low levels in the brain [100]. AXL is highly expressed in the lungs and trachea, where ACE2 expression is restricted to certain cell types [40], and is thought to play an important role in SARS-CoV-2 infection and pathology.

AXL promotes the replication of dengue virus (DENV), zika virus, Ebola, Lassa virus, Marburg, Hantaan virus (HTNV), and Andes virus (ANDV) [65,66,67,68,69,70,71]. The entry of zika viruses is mediated through the binding phosphatidylserine (PS) on the viral membrane to GAS6, which interacts with AXL, followed by viral entry into cells through endosomal pathways [65,101,102]. AXL acts as an alternative receptor for SARS-CoV-2 [20,40]. Wang et al. identified AXL by analyzing SARS-CoV-2 spike binding proteins from cells that do not express ACE2 (lung-derived cell line NCI-H1299 and bronchus-derived cell line BEAS-2B) using proteomic approaches [40]. The role of AXL in SARS-CoV-2 entry was established in three independent approaches: knocking down AXL in ACE2-negative AXL-high cells, which blocked infection; using soluble AXL to inhibit viral infection, and introducing AXL to AXL-negative HEK293T cells which enabled infection [40]. Co-immunoprecipitation and western blot analysis showed that the NTD of the spike protein but not the RBD protein interacted with AXL [40].

Bohan et al. showed that co-transfection of ACE2 and AXL or other PS receptors (Tim1 and Tim4) promotes ACE2-dependent SARS-CoV-2 infection [20]. E-64, a cysteine protease inhibitor that blocks endosomal cathepsin activity, suppresses ACE2-dependent viral entry as well as AXL or Tim1-mediated enhancement of viral entry in the presence of ACE2 [20]. Interestingly, the transfection of AXL or Tim1 does not promote ACE2-mediated viral entry in cells co-transfected with TMPRSS2, and infection was insensitive to cysteine protease inhibitor E-64, indicating that PS receptors are not required for plasma membrane-mediated infection [20]. The AXL-specific inhibitor bemcentinib inhibits SARS-CoV-2 infection in various lung cell lines, including some with low ACE2 but does not inhibit infection of Calu-3 cells, which express high levels of TMPRSS2. In contrast to the report by Wang et al. [40], reports by other investigators showed that HEK293T cells expressing AXL alone (without ACE2) do not support SARS-CoV-2 infection [20,40]. Additionally, flow cytometric studies found no binding of NTD-Fc fusion protein to HEK293T cells expressing AXL. These findings were confirmed by a lack of binding of purified NTD to recombinant AXL in biolayer interferometry studies [20]. Instead, AXL or Tim1 bind to virion-associated PS. It is not clear whether the AXL ligand GAS6 is involved in viral entry into cells in the absence of high expression of ACE2 and AXL.

## 4. C-Type Lectins

CD209L/L-SIGN/CLEC4M, CD209/DC-SIGN/CLEC4L, and CLEC4G/LESCtin are C-type lectins. These receptors function as cell adhesion molecules and pathogen receptors; they play an important role in immune responses [103,104,105,106,107]. CD209L expression is abundant in AT2 cells and endothelial cells of the lung, liver, renal vessels, and lymph nodes, whereas CD209 is mainly expressed in dendritic cells and macrophages [104]. CD209L and CD209 act as receptors for a broad range of viruses, including Sindbis virus, Ebola virus, Japanese encephalitis virus, HIV, HCV, Influenza A virus, and SARS-CoV [71,72,73,74,75,76,77,78,79]. Recent studies have indicated that CD209L and CD209 are involved in ACE2-independent SARS-CoV-2 viral entry [41]. Amraei et al. showed productive SARS-CoV-2 infection of human umbilical endothelial cells immortalized with telomerase (HUVEC-TERT cells), which express CD209L but not ACE2 [41]. The addition of soluble CD209L or CD209L knockdown significantly reduced the viral entry of pseudotyped viruses expressing SARS-CoV-2 spike proteins. The overexpression of CD209L or CD209 supported the infectivity of pseudotyped SARS-CoV-2 viruses in HEK293T cells. Spike RBD-Fc-myc or RBD-His tagged proteins bind to CD209L from lysates of HEK293T cells overexpressing CD209L or from lysates of HUVEC-TERT cells [41]. Gu et al. showed that CD209L binds to the spike NTD, RBD, and S2 regions, with the highest affinity for the NTD [42]. Hoffmann et al. demonstrated the binding of CD209 to trimeric SARS-CoV-2 spike proteins using ELISA, surface plasmon resonance, and high-speed atomic force microscopy [43]. Interestingly, CD209L interacts with ACE2, but the interaction does not involve the C-type lectin domain, which functions as a calcium-dependent glycan-recognition domain [43]. The enzymatic removal of high-mannose N-linked glycans from CD209L enhances the binding of CD209L to spike RBD-His tagged proteins [41]. Although CD209L and CD209 genes share more than 86% homology, their expression pattern is distinct. The role of CD209 in SARS-CoV-2 entry in myeloid cells (macrophages or dendritic cells) remains to be determined.

Two independent groups identified CLEC4G/LSECtin as SARS-CoV-2 receptors using cell-based library screening approaches [43,44]. Zhu et al. used a genome-wide CRISPR activation gain-of-function screen to identify a number of novel host factors that facilitate SARS-CoV-2 infection [44]. CLEC4G is one of three validated functional receptors for SARS-CoV-2. CLEC4G is expressed in sinusoidal endothelial cells of the liver, lymph node, human peripheral blood, thymic dendritic cells, and monocyte-derived macrophages and dendritic cells [106]. CLEC4G modulates T cell immune responses [107,108], interacts with filovirus glycoproteins and SARS-CoV spike proteins [80], and acts as a receptor for Japanese encephalitis virus, Marburg, Lassa, and Ebola virus [71,74,81,82,83]. CLEC4G specifically interacts with N-acetyl-glucosamine but not mannan- or N-acetyl-galactosamine–containing matrices [106]. Zhu et al. showed that CLEC4G binds to the NTD of SARS-CoV-2 spike proteins. CLEC4G was identified as a SARS-CoV-2 receptor by introducing CLEC4G into HEK293T cells and by knocking down CLEC4G in SH-SY5Y cells [44]. Soluble CLEC4G was shown to have a moderate inhibitory effect on SARS-CoV-2 infection in Huh7.5 cells [44].

Hoffman et al. used a comprehensive library of mammalian carbohydrate-binding proteins (lectins) to probe critical sugar residues on the full-length trimeric spike and the RBD of SARS-CoV-2 [43]. Annotated mouse 143 lectin-carbohydrate recognition domains were cloned and expressed as IgG2a-Fc fusion proteins from human HEK293-F cells. Using glycosylated monomeric RBD and full-length trimeric pre-fusion SAR-SoV-2 spike proteins expressed from human HEK293-6E cells to screen the carbohydrate recognition domain library, mouse Clec4g (LSECtin) and CD209c (mouse CD209, SIGNR2) were identified as high-affinity binding proteins. The addition of N-glycans (by PNGase F treatment) to spike proteins reduced their binding to mouse Clec4g and CD209c. High-speed atomic force microscopy showed that multiple lectin proteins bind to one SARS-CoV-2 spike trimer at an average density of 3.5 human CLEC4G and 3.6 human CD209 fusion molecules per trimeric spike. Human and murine CLEC4G specifically bind to N-glycans with an unsubstituted GlcNAcβ-1,2Manα-1,3Man arm. CD209c recognized all N-glycan structures that displayed terminal unsubstituted GlcNAc residues. Structure modeling showed that the N-glycan at N343, which is located within the RBD, is the spike glycosylation site most abundantly decorated with terminal GlcNAc and interacts with human CLEC4G. Unlike CD209, the binding of CLEC4G to the spike interferes with the ACE2/RBD interaction. High concentrations of murine Clec4g, murine CD209c, and human CLEC4G but not of ASGR1 reduced SARS-CoV-2 infection of Vero and Calu-3 cells. Human CLEC4G and CD209 have been demonstrated to act as ACE-independent SARS-CoV-2 receptors [41,44], but the functions of CD209c and mouse Clec4g as alternative receptors in cells without ACE2 remain to be determined.

## 5. LDLRAD3 and TMEM30A

LDLRAD3 and TMEM30A were also identified as ACE2-independent SARS-CoV-2 receptors through the genome-wide CRISPR activation screen [44]. LDLRAD3 is expressed in various tissues, including the brain, respiratory system, GI tract, reproductive tracts, and connective and soft tissues; its expression is abundant in myeloid cells [109] LDLRAD3 promotes the activity of E3 ubiquitin ligases [110] and serves a receptor for Venezuelan equine encephalitis virus, a neurotropic alphavirus transmitted by mosquitoes that causes encephalitis and death in humans [88]. TMEM30A (CDC50A) is expressed in all tissues and various types of cells [111]. TMEM30A is the beta-subunit of the phospholipid flippase (P4-ATPase), which regulates the translocation of PS from the outer to the inner leaflet of the plasma membrane, maintaining an asymmetric distribution of the phospholipid and ‘eat-me’ signal recognized by macrophages [112,113,114]. TMEM30A plays a role in the survival of hematopoietic cells [115], and TMEM30A gene knockdown improves chemotherapy treatment outcomes in diffuse large B-cell lymphoma [116]. TMEM30A, together with NRP2, and CD63, is involved in the cellular entry of the Lujo virus, an arenavirus that causes fatal hemorrhagic disease in humans [89]. The role of TMEM30A in Lujo viral entry has been demonstrated in HAP1 cells using knockdown and gain-of-function approaches but has not been confirmed in primary human umbilical vein endothelial cells [89].

Zhu et al. showed that, similar to CLEC4G, both LDLRAD3 and TMEM30A bind to the NTD of SARS-CoV-2 spike proteins [44]. The function of LDLRAD3 and TMEM30A as ACE2-independent SARS-CoV-2 receptors was demonstrated by the expression of these molecules in HEK293T cells and by loss-of-function assays in various cell types. Additionally, soluble LDLRAD3 was found to inhibit SARS-CoV-2 infection [44].

## 6. KREMEN1 and ASGR1

Gu et al. identified several SARS-CoV-2 binding proteins using a high-throughput receptor profiling system. After screening 5054 human membrane proteins for interaction with the SARS-CoV-2 extracellular spike and Fc fusion protein, KREMEN1 and ASGR1 were found to serve as ACE2-independent SARS-CoV-2 receptors [42].

KREMEN1 is expressed in various tissues, including the brain, esophagus, endocrine and reproductive tissues, and skin and is a negative regulator of Wnt signaling [117,118]. KREMEN1 also controls cell survival in a Wnt signaling-independent manner [119] and is an entry factor for coxsackievirus A10 and other human-type A enteroviruses [84,85,86].

Asialoglycoprotein receptor-1 (ASGR1, CLEC4H1) is a calcium-dependent C-type lectin receptor and is expressed primarily in hepatocytes [120]. ASGR1 internalizes asialoglycoproteins after the removal of the terminal sialic acid of the attached glycans [120]. ASGR1 is a receptor for the hepatitis E virus (HEV), and viral entry is mediated through the binding of ASGR1 to HEV ORF2 [87]. The surface asialoglycoprotein receptor ligands asialofetuin, asialoganglioside, and fibronectin competitively inhibit the binding of HEV to hepatocytes in the presence of calcium [87].

KREMEN1 and ASGR1 interact with SARS-CoV-2 spike proteins but not with SARS-CoV spike proteins [42]. Both receptors bind with high affinity to spike RBD but also interact with the NTD. ACE2-independent viral entry via KREMEN1 and ASGR1 was demonstrated in cells without ACE2 and in mice [42]. Additionally, ACE2-neutralizing Abs suppress SARS-CoV-2 entry in Calu-3, Calu-1, and Huh-7 cells, all of which express high abundance ACE2, but ACE2-neutralizing Abs do not block viral entry in HTB182 and Li-7 cells. Receptor knockdown analyses and antibodies against KREMEN1 and ASGR1 showed that SARS-CoV-2 enters HTB182 cells via KREMEN1 and Li-7 cells via ASGR1. Both ACE2 and KREMEN1 contribute significantly to viral entry in NCI-H1944 and NCI-H661 cells [42].

## 7. Conclusions and Perspective

ACE2 is an important and well-studied receptor for SARS-CoV-2 infection, but its distribution cannot explain SARS-CoV-2-mediated pathology. A number of alternative ACE2-independent receptors with broader distribution patterns may contribute to SARS-CoV-2 infection and pathogenesis. Indeed, the effects of alternative receptors on lung pathology have been demonstrated in SARS-CoV-2-infected mice. The molecular mechanisms of SARS-CoV-2 entry via alternative receptors are not always consistent and require further investigation (Figure 1). Most studies have relied heavily on the overexpression of alternative receptors in HEK293T cells or on the use of the spike proteins that are fused to other molecules (e.g., Fc). Studies using pseudotyped viruses expressing site-specific mutations of spike proteins or using monoclonal antibodies against site-specific regions of spike proteins will provide insights into the binding of SARS-CoV-2 viruses to alternative receptors in relevant cell types, particularly in primary cells and tissues. Spike mutants that are resistant to monoclonal antibodies against RBD [121] would be useful tools for identifying viral determinants of alternative receptors. Spike mutation E484D results in ACE2-independent viral entry [31], but the specific alternative receptor for this mutant, as well as for other variants that are less dependent on ACE2, have not been identified. It is not clear whether alternative receptor-mediated viral infection impacts ACE2-mediated infection or immune responses, as some receptors may either interfere with or promote ACE2-dependent infection. Elucidating the various ways that individual alternative receptors acting alone or in concert with ACE2 or with other alternative receptors impact virus-mediated immune responses, immune escape, pathogenesis, and disease progression will provide insight and strategies to help develop new therapeutics for the prevention and treatment of SARS-CoV-2 infection.

## Figures and Tables

**Figure 1 viruses-14-02535-f001:**
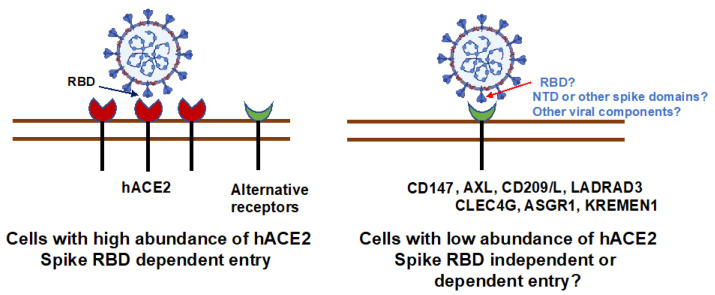
ACE2-dependent and independent SARS-CoV-2 viral entry. SARS-CoV-2 may preferentially use ACE2 for viral entry in cells with high abundance of ACE2. The ACE2-dependent virus entry is mediated through the spike RBD. In cells with low abundance of ACE2, SARS-CoV-2 can enter cells via various alternative receptors, but the entry mechanism remains to be defined.

**Table 1 viruses-14-02535-t001:** ACE2-independent alternative receptors and viral components that interact with receptors.

Receptor	Viral Components	Reference
CD147	NA, anti-RBD mAb resistant	[30]
	No full-length spike or RBD binding	[37,38]
	RBD	[39]
AXL	PS in virions (ACE2 dependent)	[20]
	NTD	[40]
CD209L/L-SIGN/CLEC4M	RBD (N-glycans)	[41]
	NTD (high affinity), RBD, S2	[42]
CD209/DCSIGN/CLEC4L	spike trimer	[43]
CLEC4G/LSECtin	RBD (N-glycans)	[43]
	NTD	[44]
KREMEN1	RBD (high affinity), NTD	[42]
ASGR1/CLEC4H1	RBD (high affinity), NTD	[42]
LDLRAD3	NTD	[44]
TMEM30A/CD50A	NTD	[44]
Clec4g (mouse)	RBD (N-glycans)	[43]
CD209c (mouse)	RBD (N-glycans)	[43]

RBD: receptor binding domain. NTD: N-terminal domain. PS: phosphatidylserine. NA: not available.

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
