# Peer review of "ACE2-Independent Alternative Receptors for SARS-CoV-2"

_viruses, 2022, doi:10.3390/v14112535_

Round 1
Reviewer 1 Report
The review manuscript by Zhang and colleagues have summarized the recent research developments on the SARS-CoV-2 entry receptors. This is nice and short review paper, and I am sure other investigators will find it informative. However, I would recommend the followings.
1. The authors in the abstract say ACE2 state “ACE2 plays a dominant role in SARS-CoV-2 replication”. I am not sure how the authors came to this conclusion. Surely, ACE2 is extensively studied in SARS-CoV2 entry but whether it plays a dominant role compared to other entry receptors yet to be established. To my knowledge there is no comparative study involving ACE2 and other entry receptor on viral entry. If authors think their statement is accurate, they need to provide additional support for their claim.
2. Authors have totally ignored the role of co-receptors in viral entry such as vimentin and heparin sulfates, etc. This needs to be discussed.
3. Moreover, a cartoon of receptors involved in SARS-CoV-2 entry with their key domain information such as Ig, and CAR domains will be helpful.
Author Response
We thank the constructive comments from the reviewer #1. The followings are our responses.
- We thank the reviewer's insight and have revised the manuscript in response to the reviewer's suggestion.
- Vimentin and heparin sulfates are ACE2 co-receptor for SARS-CoV-2. In this manuscript, we focused on ACE2-independent alternative receptors. In the revised manuscript, we included vimentin in the introduction paragraph. Heparin sulfate was mentioned.
- We include a cartoon of ACE2 and alternative receptors involved in SARS-CoV-2 entry. The spike domains and viral components involved in ACE2-independent alternative receptor are poorly characterized. Most studies are based on recombinant proteins.
Reviewer 2 Report
In this review, Zhang, Lim & Chang provided a comprehensive summary of the alternative receptors on host cells that are utilized by SARS-CoV-2 for viral entry, independent of ACE2. By introducing related works on these alternative receptors including CD147, AXL, C-type lectins, LDLRAD3/TMEM30A, and KREMEN1/ASGR1, the authors provided the readers with a detailed knowledge of the field, including the methods/platforms applied in different studies, inconsistencies between research groups and possible reasons, as well as potential future directions. The manuscript is well-written and presents the conclusions clearly, it would be great if the following minor points could be addressed:
· Table 1 is a very good summary of the alternative receptors in this review. I’m wondering whether the authors could add two more columns to include more information within the summary table. One column may indicate which cell types and which tissues are these receptors highly expressed. Another column may indicate which other viruses have been studied to utilize each specific receptor.
· In line 60, the authors mentioned that the ACE2-independent SARS-CoV-2 entry is often resistant to antibodies targeting the spike RBD, this may need more supporting material. Also the authors may add some content about the variants of concerns as well as other related sarbecoviruses.
· For the receptors binding to RBD, is the receptor-binding motif (RBM) not involved at all? The conclusion may be revealed by the neutralization effects by neutralizing antibodies targeting different regions of RBD from different directions, and whether those effects were directly through steric hindrance will need to be investigated.
· There are multiple typos found within the manuscript:
In Table1, “spike trimmer” should be “spike trimer”;
In line 258, “bine with” should be “bind with”;
In line 270, “infecntion” should be “infection”; …
The authors may need to go through and double check the spelling in the manuscript.
· Small formatting issues:
The very last column of Table 1 for the Reference may need to be aligned;
In line 197 and line 211, the first letter of the word “Spike” doesn’t need to be capitalized;
In line 198, the full name of RBD “receptor binding domain” doesn’t need to be shown again since it has appeared before.
Author Response
We thank the reviewer's for the helpful comments. The followings are point-by-point responses.
- The reviewer suggested to include additional columns for cell types and viruses. We have include a new table listing the involvement of alternative receptors in viruses other than SARS-CoV-2. We are not comfortable to include the cell types and tissues expressing these alternative receptors because 1) published papers are not always designed to characterize the expression of these receptors in various cell types and tissues so the list may be biased toward reported cell types and tissues, and 2) the information regarding expression of some receptors was found in the proteinatlas website, which will need to be validated further.
- In line 60, the authors mentioned that the ACE2-independent SARS-CoV-2 entry is often resistant to antibodies targeting the spike RBD, this may need more supporting material. Also the authors may add some content about the variants of concerns as well as other related sarbecoviruses. We have cited the study supporting that CD147-mediated SARS-CoV-2 entry is resistant to anti-RBD Ab. (see ref 30 in the revised manuscript). Most studies on alternative receptors do not test the sensitivity to anti-RBD or NTD Ab or variants. We have discussed this aspect in the introduction and conclusion, and hope to encourage readers to investigate the potential role of alternative receptors in SARS-CoV-2 variants under the immune pressure of anti-RBD Ab in vaccinated subjects.
- For the receptors binding to RBD, is the receptor-binding motif (RBM) not involved at all? The conclusion may be revealed by the neutralization effects by neutralizing antibodies targeting different regions of RBD from different directions, and whether those effects were directly through steric hindrance will need to be investigated. The anti-RBD Ab used in our study (ref 30 for CD147) did bind to RBM but the region outside of RBM is needed for the neutralization of SARS-CoV-2 in ACE2 expressing cells. Further investigation on the binding of the specific region of spike protein to alternative receptors is needed, which was discussed in the conclusion.
- Regarding typo and editorial errors, we thank the reviewer for paying attention to the details. We have made corrections and used the journal template for our revised manuscript to avoid the cosmetic error.
Round 2
Reviewer 1 Report
No further comments.